# Prevalence of elevated liver transaminases and their relationship with alcohol use in people living with HIV on anti-retroviral therapy in Uganda

J. Morgan Freiman[1¤], Robin Fatch[2], Debbie Cheng[3], Nneka Emenyonu[2], Christine Ngabirano[4], Carolina Geadas[1], Julian Adong[4], Winnie R. Muyindike[4,5], Benjamin P. Linas[1], Karen R. Jacobson[1], Judith A. Hahn[2,6]*

1 Department of Medicine and Section of Infectious Diseases, Boston Medical Center, Boston, MA, United States of America, 2 Department of Medicine, University of California, San Francisco, CA, United States of America, 3 Department of Biostatistics, Boston University School of Public Health, Boston, MA, United States of America, 4 Department of Medicine, Mbarara University of Science and Technology, Mbarara, Uganda, 5 Mbarara Regional Referral Hospital, Mbarara, Uganda, 6 Department of Epidemiology, University of California, San Francisco, San Francisco, CA, United States of America

¤ Current address: Department of Medicine and Section of Infectious Diseases, Lahey Hospital and Medical Center, Burlington, MA, United States of America
* Judy.hahn@ucsf.edu

**Data Availability Statement:** Hahn, Judith (2021), Prevalence of elevated liver transaminases and their relationship with alcohol use in people living

## Abstract

### Background

Isoniazid preventive therapy (IPT) reduces tuberculosis reactivation and mortality among persons living with HIV (PLWH), yet hepatotoxicity concerns exclude "regular and heavy alcohol drinkers" from IPT. We aimed to determine the prevalence of elevated liver transaminases among PLWH on antiretroviral therapy (ART) who engage in alcohol use.

### Setting

The Immune Suppression Syndrome Clinic of Mbarara, Uganda.

### Methods

We defined elevated liver transaminases as $\geq$1.25 times (X) the upper limit of normal (ULN) for alanine aminotransferase (ALT) and/or aspartate aminotransferase (AST). We evaluated the associations of current alcohol use and other variables of interest (sex, body mass index, and ART regimen) with elevated transaminases at study screening, using multivariable logistic regression to obtain adjusted odds ratios (aOR) and 95% confidence intervals (CI).

### Results

Among 1301 participants (53% female, median age 39 years, 67.4% current alcohol use), 18.8% (95% CI: 16.8–21.1) had elevated transaminases pre-IPT, with few (1.1%) severe ($\geq$5X the ULN). The proportion with any elevation among those currently using alcohol and

with HIV on anti-retroviral therapy in Uganda, Dryad, Dataset, https://doi.org/10.7272/Q66H4FPZ.

**Funding:** We received funding from the National Institute on Alcohol Abuse and Alcoholism U01 AA020776 (Judith A. Hahn), K24 AA022586 (Judith A. Hahn), and U24 AA020779 (Debbie Cheng), and the National Institute of Allergy and Infectious Disease P30 AI042853 (Debbie Cheng), and R01 AI119037 (Karen R Jacobson). The funders had no role in study design, data collection and analysis, decision to publish, or preparation of the manuscript.

**Competing interests:** Dr. Debbie Cheng serves on Data Safety and Monitoring Boards for Janssen. Her work for Janssen is unrelated to the submitted work and Janssen had no role in any aspect of this work. This does not alter our adherence to PLOS ONE policies on sharing data and materials.

those abstaining was 22.3% and 11.6%, respectively (p<0.01). In multivariable analyses, those currently using alcohol had higher odds of elevated transaminases compared to those abstaining (aOR 1.65, 95% CI 1.15–2.37) as did males compared to females (aOR 2.68, 95% CI 1.90–3.78).

## Conclusions

Pre-IPT elevated transaminases among PLWH receiving ART were common, similar to prior estimates, but severe elevations were rare. Current drinking and male sex were independently associated with elevated transaminases. Further research is needed to determine the implications of such transaminase elevations and alcohol use on providing IPT.

## Introduction

Tuberculosis (TB) is the leading cause of mortality for people living with HIV (PLWH) worldwide accounting for nearly one-third of all HIV deaths [1]. Isoniazid preventive therapy (IPT) is known to reduce mortality in PLWH and is currently recommended along with anti-retroviral therapy (ART) for those without active TB disease [2]. However, "regular and heavy alcohol consumption" is listed as a contraindication of IPT. This is concerning, as heavy alcohol consumption is associated with a three-fold increase in risk of TB disease [3] and with slower treatment response and higher mortality [4–7]. We previously showed, through a Markov simulation model, that the benefits of 6 months of IPT outweigh the toxicity risks in PLWH who drink alcohol when given in high TB burden settings [8]. However, this model was based on limited data on the rate of liver toxicity in individuals receiving IPT.

The risk of IPT hepatotoxicity among PLWH on ART is not well established. A previous study reported nearly a 4-fold increase in hepatotoxicity risk among severely immunosuppressed PLWH who had baseline liver transaminase elevations ($\geq$1.25 the upper limit of normal [ULN]) in alanine aminotransferase (ALT) or asparatate aminotransferase (AST) prior to starting IPT [9]. Thus pre-IPT transaminase elevations are important to consider in populations that are candidates for IPT. Transaminase elevation prevalence is believed to be 10–22% in the general population [10] and 13–43% of PLWH, independent of ART status [9, 11–16]. Alcohol is known to be hepatotoxic, and data from studies of PLWH in the US suggest heavy alcohol use is associated with increased transaminases [17, 18]. Yet the proportion of elevated transaminases among PLWH who use alcohol in TB endemic settings has not been established. Other observed contributors to transaminase elevations include ART use, male sex, viral co-infections (hepatitis B and C) [11], and liver diseases such as non-alcoholic fatty liver disease (NAFLD) [12, 13]. Thus, we evaluated the prevalence of elevated liver transaminases in PLWH on ART in Uganda and whether factors including current alcohol use (the main exposure of interest), sex, ART regimen, or body mass index (BMI) (an indicator of NAFLD), were associated with pre-IPT transaminase elevations.

## Methods

### Study population

The Alcohol Drinker's Exposure to Preventive Therapy for Tuberculosis (ADEPTT) cohort study is a one-arm trial of IPT among PLWH with latent TB infection at the Immune Suppression Syndrome (ISS) clinic of the Mbarara Regional Referral Hospital (MRRH), Mbarara,

Uganda (NCT NCT03302299). The sample for this study was persons undergoing intake-screening for the ADEPTT study. Persons were referred to the study by clinic staff and were determined eligible for transaminase screening if they were adults ≥18 years of age, fluent in English or Runyakole (the local language), HIV-infected, currently taking ART (>6 months), and living within 2 hours of the ISS clinic. Pregnant women, participants on nevirapine, those with history of active TB or taking TB medications, those on anti-convulsive medications, and those living more than 2 hours from the clinic or with plans to move, were excluded. The ADEPTT study aims to examine IPT hepatoxicity among those currently using alcohol and those abstaining in a 2 to 1 ratio, therefore the screening sample is also enriched for self-reported current (prior 3 months) alcohol use. Participants who reported abstaining from alcohol for at least the past year were included as abstainers, while those who reported alcohol use within the past year but not in the past 3 months were excluded. The study activities were approved by the Ethics review boards of Mbarara University of Science and Technology, University of California San Francisco, Boston University, and Boston Medical Center. All participants gave written informed consent.

## Outcome variable

We conducted ALT and AST testing, measured in units per liter, using venous blood samples at the MRRH ISS Clinic laboratory. We defined elevated transaminases as ≥1.25 times (X) the ULN for either ALT or AST (or both). We also calculated elevation grade, using the greater of the ALT and AST values, with Grade 1 defined as 1.25- <2.5X the ULN, Grade 2 defined as 2.5- <5X the ULN, Grade 3 defined as 5- <10X the ULN, and Grade 4 defined as ≥10X the ULN, per the Division of AIDS definitions [19].

## Independent variables

Current alcohol use, the main independent variable, was measured at ADEPTT study screening as in the prior 3 months versus past (>1 year) or never (S1 File). We also assessed sex, and age. Height and weight were retrieved from the electronic medical records of the ISS clinic to calculate BMI. BMI was categorized as "underweight" (<18.5), "normal weight" (18.5 - <25), "overweight" (25 - <30), "obese" (≥30). ART regimen at study screening, or at the clinic visit prior to screening, was also retrieved from clinic records; ART regimen was defined as a 3-category variable: "Dolutegravir-based", "non-nucleotide reverse trascriptase inhibitor (NNRTI)-based", or "protease inhibitor (PI)-based". A subset of participants had participated in previous alcohol/HIV research at the ISS Clinic in which unhealthy alcohol use was measured [20, 21]. This measure was a combination of self-reported alcohol use, using the Alcohol Use Disorders Identification Test–Consumption (AUDIT-C, modified to cover the prior 3 months) and the alcohol biomarker phosphatidylethanol (PEth), a phospholipid that forms only in the presence of alcohol [22]. We combined self-report with PEth to overcome previously noted under-reporting in this population [23]. The definition of prior unhealthy alcohol use was AUDIT-C positive (≥3 for women and ≥4 for men), and/or PEth result ≥50 ng/mL at any prior study visit.

## Statistical analyses

We estimated the prevalence of transaminase elevations overall and by current alcohol use, sex, BMI, and ART regimen. To examine associations between independent variables, i.e. current alcohol use (main exposure of interest), BMI, sex, and ART regimen, and the outcome, transaminase elevations, we used multiple logistic regression models and calculated odds ratios (OR) and 95% confidence intervals (CI), including the 4 independent variables of interest (i.e.

current alcohol use at screening, sex, BMI, and ART regimen) and controlling for age as a potential confounder. We conducted an additional exploratory analysis among those recruited from our prior research, to explore prior unhealthy alcohol use. For this model, we replaced current alcohol use at screening with prior unhealthy alcohol use as the predictor of interest. We used multiple imputation (using 25 imputed datasets using the above variables) to impute height values because height was missing from 86 ISS clinic records. The multivariable results are using the imputed data. All analyses were performed using Stata 14.2.

## Results

There were 1301 persons tested for transaminate elevations during ADEPTT study screening, from May 2018 through January 2020. The mean age was 40 years (standard deviation 9.8), and more than half were female (53.0%) (Table 1). More than one-third of the participants with BMI data were overweight (37.3%) with a BMI ≥25. Nearly two-thirds reported current alcohol use (n = 877, 67.4%), by design. Nearly three-fourths of participants were on a NNRTI-based ART regimen (n = 960, 73.8%), with 17.9% (n = 233) on a Dolutegravir-based regimen and 8.3% (n = 108) on a PI-based regimen. AUDIT-C and/or PEth data were available from our prior research for 330 participants, and 175 (53.0%) were categorized as having prior unhealthy alcohol use.

In the total sample (n = 1301), 245 participants (18.8%; 95% CI: 16.8–21.1) had any transaminase elevations. Of the 245 with any elevations, 192 (78.4%) were Grade 1 (1.25- <2.5X the ULN), 39 (15.9%) were Grade 2 (2.5- <5X the ULN), 12 (4.9%) were Grade 3 (5- <10X the ULN), and 2 (0.8%) were Grade 4 (≥10X the ULN); the prevalence of a Grade 3 or 4 elevation overall was 1.1%. The proportion with any transaminase elevations among those with current alcohol use was 22.3% (95% CI: 19.7%-25.2%), compared to 11.6% (95% CI: 8.8%-15.0%) among those abstaining (Table 2).

In multivariable analyses adjusting for the independent variables of interest plus age (Table 2), those with current alcohol use had 1.65 times the odds of any transaminase elevations compared to those abstaining (95% CI: 1.15–2.37) and males had an increased odds of transaminase elevations compared to females (adjusted odds ratio (aOR) 2.68, 95% CI: 1.90–3.78). Compared to NNRTI-based regimen, those on a PI-based regimen had a decreased odds of elevations (aOR 0.13, 95% CI: 0.05–0.36). There was no evidence of an association between BMI or age and transaminase elevations.

In sensitivity analyses of transaminase elevations limited to prior study participants (n = 330), the aOR for prior unhealthy alcohol use was 1.63 (95% CI: 0.88–3.05), after adjusting for sex, BMI, ART, and age.

## Discussion

PLWH who drink alcohol are at a particularly high risk for TB morbidity and mortality, yet current guidelines exclude persons with heavy alcohol use from IPT due to hepatotoxicity concerns. In this Ugandan cohort of PLWH receiving ART enriched for persons consuming alcohol, the proportion of individuals with pre-IPT transaminase elevations, i.e. those ≥1.25X the ULN, (19%) consistent with previous studies of PLWH which ranged from 13–43% [9, 11–16]. In addition, we observed very few (1%) transaminase elevations in the severe range, i.e. at or above 5X the ULN, consistent with other studies of PLWH that reported fewer than 2% exhibiting such elevations [12–16].

Those reporting consuming alcohol in the prior 3 months had increased odds of transaminase elevations compared to those reporting abstaining. Male sex was also significantly associated with increased odds for transaminase elevations. This is similar to what has been described in a population of persons with HIV not on ART [12], as well as in the population of

**Table 1. Participant characteristics of those screened for the ADEPTT Study, with transminase results at screening (n = 1301).**

| Characteristic | N (%) or median (IQR) |
|---|---|
| Self-reported current alcohol use | |
| No, abstained for ≥1 year | 424 (32.6%) |
| Yes, prior 3 months | 877 (67.4%) |
| Sex | |
| Male | 612 (47.0%) |
| Female | 689 (53.0%) |
| BMI (n = 1215) | 23.4 (21.0–27.0) |
| < 18.5 | 72 (5.9%) |
| 18.5 - < 25 | 690 (56.8%) |
| 25 - < 30 | 279 (23.0%) |
| > = 30 | 174 (14.3%) |
| Age | 39 (32–46) |
| 18–35 | 465 (35.7%) |
| 36–43 | 407 (31.3%) |
| 44+ | 429 (33.0%) |
| ART regimen | |
| Dolutegravir-based | 233 (17.9%) |
| NNRTI-based | 960 (73.8%) |
| PI-based | 108 (8.3%) |
| Prior unhealthy alcohol use? (AUDIT-C hazardous and/or PEth > = 50) (n = 330) | |
| No | 155 (47.0%) |
| Yes | 175 (53.0%) |
| ALT | |
| < 1.25x ULN | 1132 (87.0%) |
| > = 1.25 - <5x ULN | 164 (12.6%) |
| > = 5 - <10x ULN | 3 (0.2%) |
| > = 10x ULN | 2 (0.2%) |
| AST | |
| < 1.25x ULN | 1103 (84.8%) |
| > = 1.25 - <5x ULN | 186 (14.3%) |
| > = 5 - <10x ULN | 11 (0.9%) |
| > = 10x ULN | 1 (0.1%) |
| ALT/AST grade (highest grade of the 2) | |
| Grade 0 (<1.25x ULN) | 1056 (81.2%) |
| Grade 1 (1.25 - <2.5x ULN) | 192 (14.8%) |
| Grade 2 (2.5 - <5x ULN) | 39 (3.0%) |
| Grade 3 (5- <10x ULN) | 12 (0.9%) |
| Grade 4 (> = 10x ULN) | 2 (0.2%) |

persons not infected with HIV. We did not detect a significant association between BMI and transaminase elevations, but persons on PI-based regimens had lower odds of transaminase elevations, consistent with the low toxicity profile of PI-based regimens. The overall low proportion of high grade elevations may be due to our exclusion of patients on nevirapine, a known hepatotoxic drug.

There are limitations to this study. First, the exclusion of patients on nevirapine may have reduced the prevalence of transaminase elevations. For those with cirrhosis or advanced viral hepatitis, the synthesis of ALT and AST may have been falsely low, though this may be rare

**Table 2. Transaminase elevations (> = 1.25x ULN) by participant characteristics, among persons screened for the ADEPTT Study (n = 1301).**

| Characteristic | n/N (%, 95% CI) with any elevations | Unadjusted OR (95% CI) | Adjusted OR* (95% CI) |
|---|---|---|---|
| Self-reported current alcohol use | | | |
| No, abstained for ≥1 year | 49/424 (11.6, 8.8–15.0) | 1.00 | 1.00 |
| Yes, prior 3 months | 196/877 (22.3, 19.7–25.2) | 2.20 (1.57, 3.09) | 1.65 (1.15, 2.37) |
| Sex | | | |
| Male | 162/612 (26.5, 23.1–30.1) | 2.63 (1.96, 3.52) | 2.68 (1.90, 3.78) |
| Female | 83/689 (12.0, 9.8–14.7) | 1.00 | 1.00 |
| BMI (n = 1215) | | | |
| < 18.5 | 20/72 (27.8, 18.5–39.5) | 1.00 | 1.00 |
| 18.5 - < 25 | 131/690 (19.0, 16.2–22.1) | 0.61 (0.35, 1.06) | 0.58 (0.33, 1.03) |
| 25 - < 30 | 54/279 (19.4, 15.1–24.4) | 0.62 (0.34, 1.13) | 0.73 (0.39, 1.36) |
| > = 30 | 23/174 (13.2, 8.9–19.2) | 0.40 (0.20, 0.78) | 0.58 (0.28, 1.19) |
| Age | | 0.94 per 10 years (0.82, 1.09) | 0.91 per 10 years (0.79, 1.06) |
| 18–35 | 87/465 (18.7, 15.4–22.5) | | |
| 36–43 | 82/407 (20.1, 16.5–24.3) | | |
| 44+ | 76/429 (17.7, 14.4–21.6) | | |
| ART regimen | | | |
| Dolutegravir-based | 49/233 (21.0, 16.2–26.8) | 1.07 (0.75, 1.52) | 0.76 (0.53, 1.13) |
| NNRTI-based | 192/960 (20.0, 17.6–22.7) | 1.00 | 1.00 |
| PI-based | 4/108 (3.7, 1.4–9.6) | 0.15 (0.06, 0.42) | 0.13 (0.05, 0.36) |
| Prior unhealthy alcohol use?** (n = 330) | | | |
| No | 20/155 (12.9, 8.4–19.2) | 1.00 | — |
| Yes | 42/175 (24.0, 18.2–31.0) | 2.13 (1.19, 3.82) | — |

*adjusted model based on multiple imputation

**AUDIT-C hazardous and/or PEth > = 50

[24]. The classification of current alcohol use was based on self-report, making it subjective to recall bias [23]. The current drinking variable did not allow us to discern between any and unhealthy drinking. However, we performed a sensitivity analysis on a subgroup of participants with a previously measured objective alcohol biomarker, and the results were similar to those for self-reported current alcohol use. We did not assess the duration of time on ART prior to intake screen or evaluate immunosuppression, which previous studies reported are risks for transaminase elevations [11, 13]. However, we did limit the sample to those on ART for at least 6 months, thereby avoiding transient elevations of treatment initiation. Further, we did not assess the concomitant use of other potentially hepatotoxic medications, remedies, or conditions. In addition, because we found few cases with serious transaminase elevations, we were unable to examine correlates of this outcome. We emphasize that this analysis focused on transaminase elevations prior to IPT initiation. Our future research from the ADEPTT Study will examine the occurrence of IPT hepatotoxicity and analyze its associations with baseline transaminase abnormalities and alcohol use.

Overall, our findings are consistent with prior studies that showed that mild liver transaminase elevations are common among PLWH, while extreme elevations are uncommon. It is unknown whether these mild elevations increase the risk for IPT hepatotoxicity in the setting of ART treatment, i.e. immunocompetence. Future research needs to address whether active alcohol use increases the risk for hepatotoxicity during IPT and, if found, to investigate strategies to mitigate risk.

## Supporting information

**S1 File. Screening form used in the ADEPTT study.**
(PDF)

## Acknowledgments

We would like to thank Mbarara University of Science and Technology, Mbarara Regional Referral Hospital, the staff of the Mbarara Regional Referral Hospital Immune Suppression Syndrome Clinic, our research and administrative teams at UCSF and MUST. We give special thanks to the study participants for their time and contribution to this work.

## Author Contributions

**Conceptualization:** Winnie R. Muyindike, Karen R. Jacobson, Judith A. Hahn.

**Formal analysis:** Robin Fatch, Debbie Cheng.

**Funding acquisition:** Judith A. Hahn.

**Methodology:** Debbie Cheng, Judith A. Hahn.

**Project administration:** Nneka Emenyonu, Christine Ngabirano.

**Resources:** Judith A. Hahn.

**Supervision:** Nneka Emenyonu, Winnie R. Muyindike, Judith A. Hahn.

**Visualization:** Robin Fatch.

**Writing – original draft:** J. Morgan Freiman, Carolina Geadas.

**Writing – review & editing:** Robin Fatch, Debbie Cheng, Nneka Emenyonu, Christine Ngabirano, Julian Adong, Winnie R. Muyindike, Benjamin P. Linas, Karen R. Jacobson, Judith A. Hahn.

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
