## [Decision Letter · Decision Letter 0]

17 Feb 2021

PONE-D-20-39571

Prevalence of Elevated Liver Transaminases and Their Relationship with Alcohol Use in People Living with HIV on Anti-Retroviral Therapy in Uganda

PLOS ONE

Dear Dr. Hahn,

Thank you for submitting your manuscript to PLOS ONE. After careful consideration, we feel that it has merit but does not fully meet PLOS ONE’s publication criteria as it currently stands. Therefore, we invite you to submit a revised version of the manuscript that addresses the points raised during the review process.

We noted that you have referred to data not shown in line 164. Please note that PLOS does not permit references to “data not shown” (http://journals.plos.org/plosone/s/data-availability). Please provide the relevant data within the manuscript, the Supporting Information files, or in a public repository. If the data are not a core part of the research study being presented, please remove any references to these data.

We look forward to receiving your revised manuscript.

Kind regards,

Sara Fuentes Perez

Staff editor

On behalf of:

Isabelle Chemin, PhD

Academic Editor

PLOS ONE

Journal Requirements:

3.Thank you for stating the following in the Competing Interests section:

"Potential conflict of interest:

Dr. Debbie Cheng serves on Data Safety and Monitoring Boards for Janssen. Her work for Janssen is unrelated to the submitted work and Janssen had no role in any aspect of this work."

5. Please include additional information regarding the survey or questionnaire used in the study and ensure that you have provided sufficient details that others could replicate the analyses. For instance, if you developed a questionnaire as part of this study and it is not under a copyright more restrictive than CC-BY, please include a copy, in both the original language and English, as Supporting Information, or include a citation if it has been published previously.

6. In the Methods, please discuss whether and how the questionnaire was validated and/or pre-tested. If these did not occur, please provide the rationale for not doing so.

Reviewers' comments:

Reviewer's Responses to Questions

**Comments to the Author**

1. Is the manuscript technically sound, and do the data support the conclusions?

Reviewer #1: Yes

2. Has the statistical analysis been performed appropriately and rigorously? 

Reviewer #1: Yes

3. Have the authors made all data underlying the findings in their manuscript fully available?

Reviewer #1: Yes

4. Is the manuscript presented in an intelligible fashion and written in standard English?

Reviewer #1: Yes

5. Review Comments to the Author

Reviewer #1: Assuming the sample is appropriate, this is a straight forward descriptive statistical application of the use of multiple logistic regression models and calculated odds ratios (OR) and 95% confidence intervals (CI), on an imputed sample. It appears that only about 7 percent of the sample had to be imputed. The results appear to follow from the analysis presentation of Table 2, in that current drinking and male sex were independently associated with elevated transaminases.

6. PLOS authors have the option to publish the peer review history of their article (what does this mean?). If published, this will include your full peer review and any attached files.

Reviewer #1: No

---

## [Author Response · Author response to Decision Letter 0]

1 Mar 2021

PONE-D-20-39571

Prevalence of Elevated Liver Transaminases and Their Relationship with Alcohol Use in People Living with HIV on Anti-Retroviral Therapy in Uganda

PLOS ONE

Dear Ms. Perez and Dr. Chemin,

Thank you for your review of out manuscript. Rebuttals to your questions are provided below.

We have done so. 

 The data are available at https://datadryad.org/stash/share/Tp491NLcUe8qTcDADYiI8RtiuIFiq9lm6RAOm9zbvBIDOI# 10.7272/Q66H4FPZ

3.Thank you for stating the following in the Competing Interests section:

"Potential conflict of interest:

Dr. Debbie Cheng serves on Data Safety and Monitoring Boards for Janssen. Her work for Janssen is unrelated to the submitted work and Janssen had no role in any aspect of this work."

This does not alter our adherence to PLOS ONE policies on sharing data and materials. 

4. We note that you have included the phrase “data not shown” in your manuscript. Unfortunately, this does not meet our data sharing requirements. 

This was in error. These data are included in the above-mentioned file. We meant to note that we were not including a separate table for the analyses, but the results were presented iwhtin the paragraph. We have removed the phrase from the manuscript. 

5. Please include additional information regarding the survey or questionnaire used in the study and ensure that you have provided sufficient details that others could replicate the analyses. For instance, if you developed a questionnaire as part of this study and it is not under a copyright more restrictive than CC-BY, please include a copy, in both the original language and English, as Supporting Information, or include a citation if it has been published previously.

We have included the screening form, that collected participant age, sex, and current alcohol use as file S1. 

6. In the Methods, please discuss whether and how the questionnaire was validated and/or pre-tested. If these did not occur, please provide the rationale for not doing so.

The only data from a questionnaire were from the standardized AUDIT-C scale, which is referenced in the text. 

Reviewers' comments:

Reviewer #1: Assuming the sample is appropriate, this is a straight forward descriptive statistical application of the use of multiple logistic regression models and calculated odds ratios (OR) and 95% confidence intervals (CI), on an imputed sample. It appears that only about 7 percent of the sample had to be imputed. The results appear to follow from the analysis presentation of Table 2, in that current drinking and male sex were independently associated with elevated transaminases.

No edits are needed in response to this comment.

---

## [Decision Letter · Decision Letter 1]

6 Apr 2021

Prevalence of Elevated Liver Transaminases and Their Relationship with Alcohol Use in People Living with HIV on Anti-Retroviral Therapy in Uganda

PONE-D-20-39571R1

Dear Dr. Hahn,

We’re pleased to inform you that your manuscript has been judged scientifically suitable for publication and will be formally accepted for publication once it meets all outstanding technical requirements.

Kind regards,

Isabelle Chemin, PhD

Academic Editor

PLOS ONE

Additional Editor Comments (optional):

Reviewers' comments:

Reviewer's Responses to Questions

**Comments to the Author**

1. If the authors have adequately addressed your comments raised in a previous round of review and you feel that this manuscript is now acceptable for publication, you may indicate that here to bypass the “Comments to the Author” section, enter your conflict of interest statement in the “Confidential to Editor” section, and submit your "Accept" recommendation.

Reviewer #1: All comments have been addressed

2. Is the manuscript technically sound, and do the data support the conclusions?

Reviewer #1: (No Response)

3. Has the statistical analysis been performed appropriately and rigorously? 

Reviewer #1: (No Response)

4. Have the authors made all data underlying the findings in their manuscript fully available?

Reviewer #1: (No Response)

5. Is the manuscript presented in an intelligible fashion and written in standard English?

Reviewer #1: (No Response)

6. Review Comments to the Author

Reviewer #1: (No Response)

7. PLOS authors have the option to publish the peer review history of their article (what does this mean?). If published, this will include your full peer review and any attached files.

Reviewer #1: No

---

## [Editor Report · Acceptance letter]

21 May 2021

PONE-D-20-39571R1 

Prevalence of elevated liver transaminases and their relationship with alcohol use in people living with HIV on anti-retroviral therapy in Uganda 

Dear Dr. Hahn:

I'm pleased to inform you that your manuscript has been deemed suitable for publication in PLOS ONE. Congratulations! Your manuscript is now with our production department. 

Kind regards, 

on behalf of

Mrs Isabelle Chemin 

Academic Editor

PLOS ONE